# The Effect of Graphene Oxide and SEBS-g-MAH Compatibilizer on Mechanical and Thermal Properties of Acrylonitrile-Butadiene-Styrene/Talc Composite

**DOI:** 10.3390/polym13183180

**Published:** 2021-09-19

**Authors:** Fatin Najwa Joynal Abedin, Hamidah Abdul Hamid, Abbas F. M. Alkarkhi, Salem S. Abu Amr, Nor Afifah Khalil, Ahmad Naim Ahmad Yahaya, Md. Sohrab Hossain, Azman Hassan, Muzafar Zulkifli

**Affiliations:** 1Green Chemistry and Sustainability Cluster, Branch Campus, Malaysian Institute of Chemical and Bioengineering, Technology University Kuala Lumpur, Taboh Naning, Alor Gajah, Melaka 78000, Malaysia; fatin.joynal@s.unikl.edu.my (F.N.J.A.); hamidah.abdhamid@unikl.edu.my (H.A.H.); nafifah.khalil@s.unikl.edu.my (N.A.K.); ahmadnaim@unikl.edu.my (A.N.A.Y.); 2Business School (UniKL BIS), University Kuala Lumpur, Kuala Lumpur 50250, Malaysia; abbas@unikl.edu.my; 3Department of Environmental Engineering, Faculty of Engineering, Karabuk University, Karabuk 78050, Turkey; salemabuamro@karabuk.edu.tr; 4School of Industrial Technology, Universiti Sains Malaysia, 11800 USM, Penang, Malaysia; sohrab@usm.my; 5School of Chemical and Energy Engineering, Faculty of Engineering, University Technology Malaysia (UTM), Skudai 81310, Malaysia; azmanh@cheme.utm.my

**Keywords:** mechanical, thermal, properties, analysis, statistics

## Abstract

In this study, acrylonitrile butadiene styrene (ABS)/talc/graphene oxide/SEBS-g-MAH (ABS/Talc/GO/SEBS-g-MAH) and acrylonitrile butadiene styrene/graphene oxide/SEBS-g-MAH (ABS/GO/SEBS-g-MAH) composites were isolated with varying graphene oxide (0.5 to 2.0 phr) as a filler and SEBS-g-MAH as a compatibilizer (4 to 8 phr), with an ABS:talc ratio of 90:10 by percentage. The influences of graphene oxide and SEBS-g-MAH loading in ABS/talc composites were determined on the mechanical and thermal properties of the composites. It was found that the incorporation of talc reduces the stiffness of composites. The analyses of mechanical and thermal properties of composites revealed that the inclusion of graphene oxide as a filler and SEBS-g-MAH as a compatibilizer in the ABS polymer matrix significantly improved the mechanical and thermal properties. ABS/talc was prepared through melt mixing to study the fusion characteristic. The mechanical properties showed an increase of 30%, 15%, and 90% in tensile strength (TS), flexural strength (FS), and flexural modulus (FM), respectively. The impact strength (IS) resulted in comparable properties to ABS, and it was better than the ABS/talc composite due to the influence of talc in the composite that stiffens and reduces the extensibility of plastic. The incorporation of GO and SEBS-g-MA also shows a relatively higher thermal stability in both composites with and without talc. The finding of the present study reveals that the graphene oxide and SEBS-g-MAH could be utilized as a filler and a compatibilizer in ABS/talc composites to enhance the thermo-mechanical stability because of the superior interfacial adhesion between the matrix and filler.

## 1. Introduction

A composite can be described as a material structure that consists of two or more distinct phases working together to attain new superior properties [1]. These phases are referred to as the matrix and dispersed phases, where the former is more ductile. The matrix phase is in a continuous form that supports and shares the load with the dispersion phase. The dispersion phase is mostly stronger than the matrix phase. Thus, it is usually identified as the reinforcing phase, which is enclosed in the matrix in a discontinuous form.

Composites are composed of various formulations, and compositions of materials act together to provide essential mechanical strength or stiffness for the composite. In general, a composite material based on a matrix can be divided into ceramic matrix composite (CMC), polymer matrix composite (PMC), and metal matrix composite (MMC) [1]. The PMC can be further classified into thermoplastic matrix composite and thermoset matrix composite. While on a reinforcement basis, composites can be categorized into a particulate composite, laminate composite, and fibrous composite, which are composed of particles, laminates, and fibers, respectively [1]. Nowadays, polymer matrix composites or polymer-based composites are widely used due to their ease of use, and low costs of materials and processing. PMC also shows exceptional properties, such as high specific stiffness, high specific strength, and good resistance from corrosion, fatigue, and abrasion. However, these qualities can only be achieved by reinforcement of polymer by a strong fibrous network as non-reinforced polymer, and have limited strength, modulus, and impact resistance. Unfortunately, despite all the advantages, PMC’s main drawbacks are that the polymers exhibit a high coefficient of thermal expansion (CTE) and have a low thermal resistance.

Polymer composites are highly demanded in engineering applications, such as automotive, electronics, aerospace, construction, and building industries, which require good mechanical properties of its composite. According to [2], the advanced polymer composites’ market size can be expected to increase by USD 4.95 billion in 2024, and the growth rate for 2020 is up to 7.37%. Increasing environmental concerns shifted the global metal industry towards the plastic industry. The types of plastics that are commonly used nowadays are polycarbonate (PC), polypropylene (PP), polyvinyl chloride (PVC), acrylonitrile butadiene styrene (ABS), and others. Furthermore, the usage of polymer composites in automotive industries offers a reduction in vehicle weight, which also increases fuel efficiency and reduces carbon dioxide emissions. For instance, due to the lightweight properties of ABS, the utilization of bumpers, wheels, door handles, mirror covers, and emblems has increased over the years.

Acrylonitrile butadiene styrene (ABS) is an amorphous polymer and is widely used in engineering thermoplastics. Initially, ABS plastic was introduced with only acrylonitrile-styrene copolymer in the 1940s. However, the limitations of properties in the copolymer led to the addition of butadiene rubber. The incorporation of the rubber phase in the acrylonitrile-styrene copolymer produced a well-balanced plastic with high impact strength and ductile properties [3]. The first industrial application of ABS was in molded parts, such as pipes and sheets. Then, in 1950, ABS was officially commercialized for domestic, textile, toys, and fashion applications when the injection molding and graft polymerization techniques were discovered [4]. Other than the above applications, ABS is also frequently used in 3D printing, using additive manufacturing technology.

ABS polymers are utilized in diverse applications because of their good thermal stability, which demonstrates high toughness and adequate stiffness. Furthermore, ABS does not easily degrade when exposed to environmental stress and has high chemical resistance properties. The most significant properties of ABS include low cost, ease of fabrication with superior surface quality and stable dimension, and low CTE. Research by [4,5] reveals that although ABS exhibits good properties, double bonds present in the butadiene rubber phase and oxidations make the polymer prone to weathering conditions and heat and light that will reduce the rigidity and surface quality of ABS. On those grounds, the ABS industry depends on molecular and morphological factors to transform the material with better properties.

Researchers and industry have initiated many approaches to enhance the mechanical and thermal properties of ABS and reduce the cost of resin. For instance, a lot of research on fillers such as talc [6,7,8,9] and graphene oxide (GO) [10,11,12,13], and compatibilizers such as SEBS-g-MA [14,15,16], has been carried out in order to develop better processability and properties of ABS at a lower cost. Talc is a versatile mineral, a hydrous magnesium silicate mineral for which its properties and composition depend on the mining location. Thus, every talc mined will have a different composition of magnesium oxide (MgO), water (H_2_O), and silicon dioxide or silica (SiO_2_) [17]. Talc is one of the important reinforcement materials in the plastic industry due to its platy structure. The structure provides talc with qualities such as high lubricity, low gas permeability, and high resistivity. Thus, the incorporation of talc mineral in resin enhances the impact strength and dimensional stability, and improves the heat distortion temperature [18].

Consequently, reinforcement of thermoplastic is highly used in industry. Filled polymer composite has gained attention due to its low cost and wide application, improving composites’ physical properties, modulus, and thermal properties. Thus, ASB and talc composites are reinforced with graphene oxide (GO) using the injection molding method for this study. Due to its great thermal, electrical, and mechanical properties, graphene oxide has attracted the researchers’ attention to deepen their knowledge on the properties and ability to enhance thermoplastics. Graphene oxide is a single monomolecular layer of graphite with several functional groups of oxygen, such as carboxyl, carbonyl, epoxides, and hydroxyl groups, which cause the composite’s unique properties. The studies conducted by [19,20] reveal that the addition of graphene oxide to the polymer shows a better dispersion of GO within the matrix and also exhibits higher solubility and possibility of surface functionalization for various applications.

Subsequently, among the studies, it is proven that introducing a compatibilizer has been an effective approach for reducing interfacial tension and improving the phase morphology of the blend [21]. In most of the studies, the compatibilization of polymer blends is carried out by the introduction of block or graft copolymers. Specifically, for this study, maleic anhydride grafted with SEBS (SEBS-g-MAH) is used with ABS as the main phase. Generally, the compatibilizer is compatible with the ABS and will react with another phase. In this case, the maleic anhydride (MAH) group will react with the hydroxyl group of graphene oxide, and the styrene of SEBS will associate with ABS. Despite all the effort in ensuring the enhancement of the matrixes using fillers and compatibilizers or other additives, it does not mean that the properties of the matrixes will definitely improve, as it is also possible that the properties are decreasing.

The demand for commodity and engineering thermoplastics is increasing every year. With this in mind, the development of thermoplastics is constantly needed to produce balance properties in terms of mechanical and thermal properties, as well as processability. In this study, ABS was used as the matrix that has high potential in widening its application area. ABS is well-known for its performance and costs, which fall between engineering and commodity plastics. As mentioned previously, the incorporation of fillers into the matrix is important for better processability and quality at a lower cost. Thus, the addition of fillers into ABS ensures that the thermoplastic competes with engineering plastic and commodity plastic in terms of performance [22]. Furthermore, particulate fillers, such as calcium carbonate, mica, Wollastonite, silica, and talc, are widely used as reinforcement in polymer composites. These fillers improve the modulus of the composite and increase the strength of the matrix in some cases. Despite the interest, very few studies are conducted on mineral fillers and ABS, and many of the researchers only focus on calcium carbonate (Ca_2_CO_3_) as a filler. Few types of research are shown on talc, which has been incorporated with PC/ABS. The study reveals that the tensile modulus increased and the CTE decreased with the addition of talc. In another study of ABS filled with talc, the results showed that the stiffness of pure ABS improved effectively, but resulted in a drastic decrease in impact strength. This is likely due to the lower interfacial adhesion between the filler and the matrix phase. The recent development of thermoplastic has led to the findings of incorporating graphene oxide (GO) in polymers because of the hydrophilicity, surface energy, and mechanical properties of the nanomaterial. However, most of the studies concentrated on the incorporation of GO with biopolymers such as poly (methyl methacrylate) (PMMA), polylactide (PLA), and poly (vinyl alcohol) (PVA) [22,23]. In particular, there are few studies that have considered using GO as a filler in ABS or ABS/talc matrix. It is expected that incorporating graphene oxide and SEBS-g-MAH in ABS/talc will result in better interfacial adhesion and improved mechanical properties and thermal stability. However, studies have been conducted on the utilization of GO or talc as fillers in ABS composites [23,24,25,26]. There are limited studies in the literature that utilize GO as a filler and SEBS-g-MAH as a compatibilizer in ABS/talc composites. Therefore, the present study was conducted to isolate ABS/talc/GO/SEBS-g-MAH and ABS/GO/SEBS-g-MAH composites with varying graphene oxide and SEBS-g-MAH loading. The objective of the present study was to increase the mechanical and thermal properties of the isolated composites. The significance of this groundwork is to develop a thermoplastic product, which is ABS with enhanced properties and is cost-effective. The addition of inorganic fillers such as GO and talc in a small amount provides the composite with good properties and can lower the cost for the industries. Other than that, the ABS composite application can be widened and replace the PC/ABS composite that is highly in demand nowadays, especially in automotive industries.

## 2. Methodology

### 2.1. Materials

Acrylonitrile-butadiene-styrene (ABS) as the matrix used in this study was a high impact strength ABS with the trade name of Toyolac 100–322, supplied by Toray Plastics (Penang, Malaysia). The properties for high impact strength ABS are presented in Table 1. Graphene oxide and silane-treated talc were used as fillers in this study. SEBS grafted with maleic anhydride (SEBS-g-MA) was selected as the compatibilizer. The process of the composite production can be seen in Figure 1.

### 2.2. Preparation of Composite

The composite will be prepared following the processes below. ABS blend formulation is shown in Table 1. The content of the graphene oxide (GO) filler and the SEBS-g-MA compatibilizer in this formulation was varied to find the optimal mechanical and thermal properties. The preparation process involved mixing ABS filled with talc/GO/SEBS-g-MA in a twin-screw extruder, followed by injection molding.

### 2.3. Melt Blending Process

ABS pellets and talc were dried in a hot-air oven at 80 °C for 4 h to remove the moisture before compounding, which caused a defect in produced ABS/talc. The filler was melt-compounded with ABS using an internal mixer (Thermo Haake 600p) at 190 °C, and a rotor speed of 90 rpm for 15 min. The stabilization of the torque value suggested the absence of further degradation of the matrix and the leveling of filler dispersion.

### 2.4. Melt Extrusion Process and Injection Molding

After the process parameters were decided, the ABS/talc were extruded with different amounts of GO and SEBS-g-MA using a twin-screw extruder. The counter-rotating screw blends and melts the materials. The twin-screw was set with a 30 rpm screw speed, and the barrel temperature to the head was set to 180, 190, 200, 210, 220, 230, and 240 °C. The GO loading in ABS/talc ranged from 0.5 to 2.0 phr, while SEBS-g-MA ranged from 4 to 6 phr. Then, the extruded pellets were crunched into small pellets using an automatic cutter attached to the extruder. Subsequently, the pellets were dried in an oven for 24 h before the injection molding process.

The extruded pellets were then injected into the injection molding machine into standard tensile, flexural, and Charpy impact samples. After 24 h of drying, the pellets were fed into the hopper of the injection molding machine with barrel temperature set from 190 to 245 °C and the mold temperature of 60 °C.

### 2.5. Mechanical Properties Test

The composites were subjected to undergo various mechanical properties tests. Ten samples of each formulation were tested, and the average value was recorded. The test specimens were initially conditioned for 48 h at 23 °C. The sample evaluated for the tensile test was performed under ambient conditions in accordance with ASTM D638 using the Lloyd Universal Testing Machine with a crosshead speed of 5 mm/min. The three-point bending flexural test was performed according to the ASTM D790 using the Lloyd Universal Testing Machine with a crosshead speed of 15 mm/min, and the support span for the flexural testing was 50 mm. The Charpy impact test was conducted according to the ASTM D256 using the Ray-Ran Pendulum Impact Tester System with a hammer weight of 1.19 kg and a speed of 2.90 m/s. Ten specimens for each formulation were cut into 8 cm length, and the thickness was determined before testing.

### 2.6. Thermal Properties Test

Thermal properties analyses of the composites were conducted using the Mettler Toledo Thermogravimetric Analyzer (TGA) model 851e, complete with nitrogen and purified air burning conditions. The samples were dried for 2 h at 100 °C before the test to minimize moisture effects. The samples were firstly placed in an alumina crucible. Then, the samples were tested with temperatures ranging from 25 to 800 °C with a heating rate of 10 °C/min under nitrogen to examine the thermal degradation behavior. The thermal degradation onset temperature and residue weight loss were determined. The infrared spectroscopy analyses were carried out using Nicolet i10s FTIR equipped with attenuated total reflection. The spectral range used was from 4000 to 500 ${\mathrm{cm}}^{-1}$ at a resolution of 2 ${\mathrm{cm}}^{-1}$ at room temperature. The data obtained were recorded in accordance with ASTM E168 in the transmittance mode.

### 2.7. Statistical Analysis

The data for different compositions were analyzed using analysis of variance (ANOVA) and descriptive statistics, including a five-number summary (boxplot), including min, max, first quartile, second quartile, and third quartile.

## 3. Results and Discussion

### 3.1. Determination of Parameters for Compounding Process

The compounding process was carried out to determine the suitable mixing parameters through several experiments. Other than the determination of parameters, the results obtained can be used to study the fusion characteristic of ABS. In the preparation procedure, the ABS was first loaded into the internal mixer and preheated for 2 min. The next 15 min of the compounding period were allocated, and the required amount of talc was added to melt ABS after 2 min. Initially, the temperature and speed of the rotor were set at 190 °C and 90 rpm to compound and allow homogenous mixing of ABS and talc. The compounding of ABS/talc was fixed to 90% of ABS and 10% of talc. In this experiment, the amount of pure ABS was 40 g, whereas, for the compounding, the amount of ABS and talc were 36 and 4 g, respectively. Table 2 outlines the parameter settings for melt mixing that were used in this study.

As ABS pellets were introduced into the compounding chamber, the initial peak was observed, indicating the increase in torque. The viscosity of the ABS reduced as it started to melt with time and pressure, where there was a decrease in torque. After 2 min, talc was added. A sharp peak was shown in the graph that indicated that the torque was decreasing, as illustrated in Figure 2. The dispersion of the sample was considered complete as the torque stabilized and showed a plateau in the graph. Talc was homogeneously blended with ABS only at a temperature of 190 °C and speed of 90 rpm. The samples differ in color, where the lower temperature has a slightly greyish color, but the color was slightly darker brownish at the higher temperature. This proved that the materials are well-preserved from being burned during the process.

The observations can be supported with the degradation characteristic of ABS and talc. [27] studied the degradation of different ABS grades: general-purpose, high heat resistance, and high-impact ABS. According to the analysis conducted, the degradation temperature for all grades ranges from 324 to 475 °C, with a one-step thermal decomposition. Moreover, the DTG curves reveal a large mass loss at a stated temperature. The analysis was supported by the results reported by the work of Jang [27], which claimed that the normal ABS degrades at 410 °C with the residue of 1% above 600 °C.

Figure 2 depicted the general fusion curve for unfilled ABS melted in Haake internal mixer, with an initial temperature of 190 °C, speed of 90 rpm, and blending time of 15 min. Referring to Figure 2, point A shows the peak for sample loading. Once the sample loading and driving force of free material flow are balanced, the torque values will decrease and generate point B. Then, torque values increase again due to the onset of fusion and compaction, which is shown in the figure as point C.

When it reaches point C, the material begins to melt at the interface between the hot surface and compacted material. This happens when the material reaches a void-free state. There is a slight increase in temperature with a long time of blending because of thermal energy absorbed by the material. In fact, increasing the temperature leads to the reduction of the material’s melt viscosity. Thus, the torque value decreases as the blending time increases. This study defines the period between fusion point A and point C as the fusion time. In contrast, the fusion percolation threshold (FPT) is the variation of torque between points B and C.

Figure 3 shows the melt mixing torque graph of unfilled ABS and ABS/talc at 30 and 90 rpm, respectively. As mentioned earlier, there was an increase in the torque value where the first sharp peak was shown in the graph, indicating the resistance in the rotor when un-melted ABS was fed into the chamber. The shearing action caused the ABS to melt and reduced the resistance towards the shearing force of the rotor. Figure 3 shows that the unfilled ABS has a shorter fusion time compared to talc-filled ABS. In addition, at a lower rotor speed of 30 rpm, the fusion time lengthened compared to unfilled ABS and talc-filled ABS at 90 rpm, respectively. Increasing fusion time can be caused by the physical characteristic of talc, which has a plate-like structure. Furthermore, the difficulty of the ABS composite to fuse with the talc filler might also occur due to the low adhesion property and low specific surface area of talc. The statement can be supported by a previous study performed by [28] on the fusion time of talc-filled PVC. The study reveals that the composite and fillers are difficult to blend, and the incorporation of talc to PVC lengthens the fusion time to two times longer than the unfilled PVC.

Table 3 illustrates the effect of adding talc filler and using different rotor speeds to blend the composites on the torque value. It can be seen that the incorporation of talc filler reduced the final torque of the ABS compound slightly due to the separation of ABS particles. Reductions of the final torque indicate reducing melt viscosity, by which less force is required to continue mixing and homogenizing.

These findings are in line with the previous finding reported by [28] on talc-filled PVC. The previous study shows that at a maintained rotor speed of 45 rpm, the fusion torque decreases with the addition of talc fillers. The study suggests that the addition of talc filler can increase the fusion temperature and reduce the melt viscosity. According to [29], addition of fillers such as SM90 and talc can increase the heat transfer and shear throughout the PVC grains. Consequently, higher shear and heat transfer offers increments in fusion temperature and a reduction of melt viscosity [28]. In addition, better heat and shear transfer between the filler and compound are promoted by talc’s characteristics, such as the orientation of particles, plate-like structure, and the surface adhesion of the talc filler. These characteristics result in friction energy between ABS and talc particles. However, in this case, the results show a small reduction in fusion torque caused by low shear and heat transfer.

Referring to Table 3, the final torque of the filled ABS exhibited a much lower torque value at an average of 7.5 Nm compared to the unfilled ABS torque value at 9.3 Nm. This proves that the addition of talc into ABS increased the processability of the composites. Therefore, for better processability and compatibility of the composites, SEBS-g-MAH and GO were incorporated in this study. Despite the limitations of this method and consequently poor results in the fusion study, our findings do, however, suggest that the fusion torque and final torque will be lower, and the processing of the composite will be easier.

### 3.2. Fusion Percolation Threshold (FPT)

Table 4 shows the effect of talc addition on the fusion percolation threshold (FPT) of ABS composites. Referring to Table 4, the incorporation of talc promotes a longer fusion time and FPT compared to unfilled ABS. These values correlate favorably with [29], and further support the concept of increasing fusion time resulting in increasing FPT of the composite. In fact, fusion time indicates the requirement of thermal energy to be absorbed in order to fuse composites and fillers. Thus, higher thermal energy than the FPT is needed to fuse the composites. The addition of talc in ABS increased the FPT to 5.5 Nm, which indicates that talc-filled ABS required higher thermal energy to be fused together.

### 3.3. Effect of Different Loading of Graphene Oxide (GO) on Tensile Strength, Young’s Modulus, and Elongation at Break of Composites

An investigation into the effect of GO and SEBS-g-MA as fillers and compatibilizers on the mechanical properties of ABS/talc composites at different loading was carried out. All the composites used were in the ratio of 90:10 by the percentage of ABS:talc ratio, as outlined in the earlier study. The amount of loading of GO and SEBS-g-MAH in the composite added ranged from 0.5 to 2.0 phr, and 4.0 to 6.0 phr, respectively. In order to investigate the properties of the composites, the mechanical properties reported are tensile strength (TS), elongation at break (EB), Young’s modulus (YM), flexural strength (FS), flexural modulus (FM), and impact strength (IM). From the findings, it was found that the addition of GO improves the interfacial adhesion between the matrix and filler particles, hence enhancing the mechanical properties of the composite. In order to study the tensile properties, tensile strength, Young’s Modulus, and elongation at break are investigated. Figure 4 shows the properties of TS, YM, and EB of ABS and ABS/talc with different loading of GO and SEBS-g-MAH. The terms ABS/talc/GO-0.5/S-6, ABS/talc/GO-1.0/S-6, ABS/talc/GO-1.5/S-6, and ABS/talc/GO-2.0/S-6 indicate the usage of 0.5, 1.0, 1.5, and 2.0 phr of GO.

From the results, it can be deduced that the incorporation of fillers into ABS increased the tensile strength of ABS composites. This indicates that the composites are ductile and strong. It is interesting to note that the tensile strengths of ABS composites with addition of GO and SEBS-g-MAH are higher than unfilled ABS but lower than ABS/talc composites. In general, properties of polymer composites are heavily dependent on the dispersion of reinforcing fillers. The dispersion will help in increasing the reinforcement surface area and affect the neighboring polymer chains. Therefore, incorporation of GO/SEBS-g-MAH with uniform dispersion onto the matrix will ensure a significant reinforcement of the composite.

Figure 4a shows the most significant increase of tensile strength using 1.5 phr of GO and 6.0 phr of SEBS-g-MA when compared to unfilled ABS. The tensile strength of the ABS/talc/GO-1.5/S-6 was higher by 11.136 MPa. Although the incorporation of filler and compatibilizer shows a reduction in TS when compared with the ABS/talc system, the formulations with 1.5 phr of GO show only a slight reduction of 3.713 MPa and can be considered as insignificant. This formulation shows a considerably good TS value compared to other formulations. The trends are also similar in Young’s modulus and elongation at break, where the YM increased by 10.3% and the EB decreased by 31% for the formulation with 1.5 phr of GO in ABS/talc. This concurs well with the previous findings of by [30], where the addition of SEBS-c-GOS into PS illustrated an increase in both tensile strength and Young’s modulus, with only 2.0 wt.% of SEBS-c-GOS. The enhancement in these properties is due to the bonding of GO’s oxygen functional group with the polar polymers. The bonding promotes a strong interfacial adhesion between the filler and matrix. Other than that, the authors also highlighted that grafting of SEBS in the composites is important for effective load transfer between the fillers and matrix.

Furthermore, the experiment was also conducted to examine the effect of GO/SEBS-g-MAH on ABS. By referring to Figure 4a–c, it can be seen that the incorporation of fillers and compatibilizers to the matrix enhanced ABS properties, especially in the strength of the composite. However, the addition does not promote a significant increment in terms of stiffness that values of YM can deduce. In a recent study of PP/GO with the addition of PP-g-MAH to the composites, the study reveals that without the addition of compatibilizer, PP and GO are incompatible with each other [13]. With the presence of PP-g-MAH, the interfacial bonding of PP/GO improved and hence resulted in better thermal properties.

### 3.4. Effect of Different Loading of SEBS-g-MAH on Tensile Strength, Young’s Modulus, and Elongation at Break of Composites

The SEBS-g-MAH is widely used as a compatibilizer in composites as it can improve the adhesion between the matrix and filler. This can promote a better interaction of particles between the matrix and filler. Thus, the effect of different loading of SEBS-g-MAH was also studied to examine the impact on the composite. The ABS/talc/GO-1.0/S-4, ABS/talc/GO-1.0/S-6, and ABS/talc/GO-1.0/S-8 indicate the usage of 4, 6, and 8 phr of SEBS-g-MAH.

Figure 5 shows an increment towards the tensile properties when compared to unfilled ABS. Compared to unfilled ABS, the composites with 6 phr of SEBS-g-MAH in the ABS/talc/GO system showed the most significantly enhanced properties, with 33% and 7% increments of TS and YM, respectively. Consequently, higher loading of SEBS-g-MAH into the system will only deteriorate the properties of the composite. For instance, adding 8 phr of SEBS-g-MAH reduced the TS and YM but increased EB extensively, up to 34%. This may be due to aggregation of reinforcing fillers and chain entanglement that cause high extensibility to reach breaking point [13]. A high amount of chemical bonds leads to chain entanglement, especially in the ABS/talc system.

By comparing the effect of SEBS-g-MAH loading in ABS/GO, the strength of composites is considerably higher compared to the addition in the ABS/talc/GO system. However, the absence of talc in the system deteriorates the stiffness of the composite. The plausible explanation is that the stiffness of the ABS composite is offered by the talc properties. Overall, it can be deduced that the higher the concentration of SEBS-g-MAH in the composite, the lower the properties of YM, but it also increases the EB.

### 3.5. Effect of Different Loading GO on Flexural Strength and Flexural Modulus

Figure 6a illustrates the flexural strength of ABS/talc/GO/SEBS-g-MAH composites. From Figure 6, it is interesting to see that the trends of flexural properties have a similar positive trend compared to tensile properties. As can be seen in Figure 6a, incorporation of GO and SEBS-g-MAH into the composite increases the flexural strength (FS) by an average of 15%. Rather than lowering the strength as depicted in tensile strength when compared to the ABS/talc system, the addition of filler and compatibilizer enhances flexural properties. It can be seen that the FS are much higher than ABS/talc and unfilled ABS composites. The flexural strength of ABS/talc/GO/SEBS-g-MAH shows a similar trend as in tensile strength, where the addition of 1.5 phr of GO has the highest increment relative to unfilled ABS and ABS/talc. The increment with 1.5 phr of GO loading in the ABS/talc system is 27.2% and 27.8% compared to ABS and ABS/talc, correspondingly. Whereas, without talc in the system, the FS is still higher compared to ABS and ABS/talc.

Figure 6b illustrates the flexural modulus of ABS/talc/GO/SEBS-g-MAH. The incorporation of all amounts of GO improved the flexural modulus in the ABS composite. Next, high FM can be seen when using 1.5 phr of GO loading when compared to ABS and ABS/talc. This is a similar trend as in FS and TS. However, without talc in the system, the modulus of the composite is lower than the ABS composite with talc filler. The existence of talc can explain this as promoting the stiffness and strength of the composite. According to Abu Bakar et al. [28] higher flexural properties are attributed to the physical characteristics of talc in terms of the orientation and particles’ aspect ratios. Thus, high flexural modulus contributed to the high resistance of bending when stress is applied.

### 3.6. Effect of Different Loading of SEBS-g-MAH on Flexural Strength and Flexural Modulus

Figure 7 presents the data on flexural strength (Figure 7a) and flexural modulus (Figure 7b) with different loading of SEBS-g-MAH. The figures demonstrated that the optimum FS and FM in both ABS/talc/GO/SEBS-g-MAH and ABS/GO/SEBS-g-MAH composites is found by using 4 phr of SEBS-g-MAH. Similar trends can be seen in YM in tensile properties, wherein the higher the amount of SEBS-g-MAH, the lower the modulus. In terms of composite stiffness, the composite with the absence of talc has a lower modulus. The addition of 4 phr of SEBS-g-MAH shows an increment of up to 31% when compared to unfilled ABS and ABS/talc. The compatibilizer also shows a higher increment of 37% when SEBS-g-MAH is added into the composite. The FM depicted in Figure 7a shows that higher loading of SEBS-g-MAH leads to a reduction of modulus properties.

### 3.7. Impact Strength of Composites with Different Loading of Graphene Oxide and SEBS-g-MAH

Figure 8a,b demonstrate the data of impact strength of the ABS composite. Talc-filled composites showed the lowest impact strength in ABS. Then, the incorporation of GO and SEBS-g-MAH showed an increment in the impact strength, but still lower compared to unfilled ABS. Furthermore, the composite without talc filler promotes higher impact strength than the talc-filled ABS, as proposed by the theory of introducing talc into a composite system. Referring to Figure 8a, the incorporation of SEBS-g-MAH shows that the higher the amount of loading, the better the impact properties. The best explanation for this matter is that the presence of GO in the composites promotes better interfacial adhesion onto the surface of the matrix and fillers. In addition, as mentioned for tensile properties, the addition of a compatibilizer is crucial to obtain better adhesion between GO and the ABS matrix.

### 3.8. Effect of Thermogravimetric Analysis (TGA) on Different Loading of GO

As mentioned earlier, ABS has become one of the common plastics to be used in engineered and commodity plastics due to its impressive properties. In addition, ABS has high sensitivity towards thermo-oxidative processes. At a range of 220–240 °C, the C=C double bond in polybutadiene undergoes cross-linking phenomena that cause the polymer to be prone to the oxidation process. ABS degradation usually happens in a single-step process. It was also reported that ABS could undergo a two-step degradation process attributed to the styrene and butadiene phases in the composite, that occurs at 425 and 650 °C, respectively [27]. However, standard TGA does not have the ability to show the separation of SAN and butadiene components. Table 5 presented the overall thermogravimetric analysis (TGA) with different amounts of GO loading in every formulation. The analysis was performed to determine the effect of graphene oxide addition into the composite. According to Abu Bakar et al. (2005) and Yuan et al. (2003) as cited in [28], the thermal stability of polymer composites can be investigated by the increase in 5% weight loss temperature.

From Table 5, the data show that the addition of GO and SEBS-g-MAH increased the temperature at 5% weight loss when compared to the ABS/talc system only. Following the trends in mechanical properties, the addition of 1.5 phr of GO also showed a positive outcome, where the temperature at 5% weight loss increased by 4.7 °C compared to ABS/talc decomposition. The incorporation of GO and SEBS-g-MAH also increased the ABS/GO/SEBS-g-MAH system when compared to both ABS and ABS/talc. Moreover, the decomposition temperature showed a higher temperature for all formulations compared to the formulations with talc. For instance, the most significant increase in thermal decomposition at 5% weight loss was shown by the addition of 2.0 phr of GO with an increment of 4.81 °C when compared to ABS.

By comparison, the presence of GO and SEBS-g-MAH improved the decomposition temperature for weight loss of 5%, 10%, 20%, and 30%. Thus, ABS/talc/GO/SEBS-g-MAH and ABS/GO/SEBS-g-MAH are more thermally stable than pure ABS and ABS/talc. In addition, retarding of emissions, dispersion of molecules, and the presence of the vast amount of oxygen functional groups are the main factors affecting the thermal stability [31]. In this case, GO might have acted as a barrier and a thermal retarding agent for the composite. Other than that, GO has a vast amount of oxygen functional groups where the exothermal energy from the decomposition of GO enhances the decomposition temperature of the ABS matrix. Similar observations have been reported by [32,33]. Ref. [33] quoted that the increase in thermal stability was attributed to the inhibition of polymer segment mobility at the interphase of the matrix and filler. This inhibition occurs because of the strong interaction between the polymer and the matrix. The determination of the inflection point can also deduce thermal degradation. In this study, the inflection point of ABS/talc/GO/SEBS-g-MAH showed a shift towards the right, when compared to the ABS/talc composite, as presented in Table 6, where the addition of fillers and compatibilizers increased the inflection point values.

For residual weight at 800 °C, ABS/GO/SEBS-g-MAH showed higher residues than ABS, ABS/talc, and ABS/talc/GO/SEBS-g-MAH. This was caused by the presence of GO in the composite.

As discussed earlier, the incorporation of SEBS-g-MAH and GO shows an increase in thermal stability. Overall, the thermal stability increased with an increasing amount of compatibilizer when incorporated with 1.0 phr of GO. These trends are similar to the trends illustrated for impact strength. The data shown in Table 6 also show that the most significant increase in decomposition temperature of ABS/talc/GO/SEBS-g-MAH was with the addition of 8 phr of compatibilizer at 2% weight loss. A similar enhancement trend can be seen in the impact strength of the composite. The addition of 8 phr of SEGS-g-MAH increased the stability by 1.75% and 2.01% compared to ABS/talc and ABS, respectively. Similar to the ABS/talc/GO/SEBS-g-MAH system, the composite without talc also contributed to higher thermal stability, with an average of 2.42% of increment at 2% of weight loss. However, in terms of inflection point temperature, the analysis did not show any significant difference between the unfilled ABS and filled ABS. However, the analysis showed a high decomposition temperature and residue with the addition of 4 phr of SEBS-g-MAH into the ABS/talc system and 6 phr into the ABS composite. However, the plausible explanation is that the interfacial adhesion between the fillers and the matrix is lower. Thus, the fillers are left as residues as the composite degrades.

### 3.9. FTIR Analysis

FTIR analysis was carried out for ABS, talc, GO, ABS/talc/GO-1.5/S-6, and ABS/GO-2.0/S-6 composites, as shown in Figure 9. FTIR analysis was performed to determine the effect of GO and SEBS-g-MAH treatment in ABS/talc un-compatibilized composites. As illustrated in Figure 9, the ABS spectrum shows peaks at 970–966, 1494, and 2237 cm^−1^, corresponding to the polybutadiene region, aromatic C-C stretching of styrene, and CN stretching, respectively. After the addition of GO and SEBS-g-MAH, there was no change in intensity of these peaks that indicates the existence of ABS in the composite. The peaks at 3448, 1637, and 1047 cm^−1^, attributed to the hydroxyl group absorbed water on the GO surface, the vibration of the un-oxidized graphitic domain, and stretching vibrations of C-OH of alcohol respectively, exist in GO [31]. Better dispersion of GO in the ABS composite can be explained by the interaction of polar groups in GO and ABS that react to form a strong hydrogen bond.

In addition, ABS/talc/GO-1.5/S6 showed new peaks at 3671 cm^−1^ that indicate the presence of the Mg-OH bond of talc in the composite. The incorporation of talc can also be seen in the IR spectrum, where the region at 1017–910 cm^−1^ shows that the intensity is higher compared with the ABS/GO/S spectrum, which indicated an interaction occurs between talc and the ABS polymer. Other than that, new peaks can be seen at 1818–1809 cm^−1^, which represent the reaction between SESB-g-MAH with GO that forms the carbonyl group at that region. Next, the peak for the isocyanate group (C=N=O) is illustrated at 2339 cm^−1^ in the ABS/GO/S system, which shows the possibility of GO-SEBS-g-MA’s interaction with the acrylonitrile region of ABS. Thus, it can be deduced that the composites are compatibilized and hence promote better interfacial adhesion between fillers and the matrix. Therefore, the mechanical properties are enhanced as discussed previously.

### 3.10. Statistical Analysis

The results of 98 experiments for various selected responses covering 14 different compositions (7 replications at each composition) are summarized and presented in a graphical form using a boxplot (Figure 10). The graphical presentation exhibited fluctuations in the behavior of selected responses to various compositions. For instance, tensile strength showed the highest response with composition ABS/Go-1.0/S-6 and the lowest response with composition ABS, whilst other compositions showed tensile strength between these two compositions.

The highest spread was exhibited with compositions ABS and ABS/Go-1.0/S-6, while other compositions exhibited a much lower spread compared to these two compositions. In general, the dispersion of the data was low and acceptable. The results of 98 experiments were further analyzed using analysis of variance (ANOVA) to investigate the effect of different compositions (14 different compositions) on 6 selected responses, namely tensile strength (TS), Young’s modulus (YM), elongation at break (EB), flexural strength (FS), flexural modulus (FM), and impact strength (IS). The results of ANOVA for the selected responses in Table 7 showed that different compositions significantly affect the six responses, which indicates that different compositions will result in different values of the selected responses. Significant results could be due to the dispersion of GO and the interfacial interactions with the matrix with the addition of SEBS-g-MAH. Additionally, talc at 10 wt.% acts as the reinforcing filler that increases the tensile modulus and stiffness but reduces the strain-to-break and impact strength. The reason for using all three is to balance the enhancement properties of the intended composite. The composition should improve the tensile and flexural strength, modulus, and strain-at-break, while maintaining the impact properties of the composite. This is evidence from the data obtained from the mechanical testing conducted. The data were further analyzed using Tukey’s test for multiple comparisons to find the composition that causes the differences, which is indicated with *p*-values less than 0.05.

## 4. Conclusions

In the present study, the influence of graphene oxide as a filler and SEBS-g-MAH as a compatibilizer was determined on the mechanical and thermal properties of ABS/talc composites. It was found that the incorporation of GO and SEBS-g-MAH yielded lower torque and eased the mixing between the matrix and fillers. The addition of GO at 0.5 to 2.0 phr showed marginal improvement of mechanical and thermal properties of ABS/talc/GO/SEBS-g-MAH and ABS/GO/SEBS-g-MAH composites. The mechanical properties of ABS/talc/GO/SEBS-g-MAH composites revealed that the inclusion of GO in the polymer matrix increased Young’s modulus, flexural modulus, and flexural strength, which might be due to better interfacial adhesion between the matrix and fillers. However, drastic reductions in impact strength are attributed to the properties of talc. In thermal properties’ analysis, it was shown that the ABS/talc/GO/SEBS-g-MAH composites needed higher temperatures to be deflected at a specific height compared to ABS and ABS/talc, which suggests the presence of GO as a thermal retardant agent. Moreover, the introduction of SEBS-g-MAH at 4 to 8 phr as a compatibilizer in the composite further increased the mechanical properties. From the mechanical and thermal properties’ analyses of the composites, the optimum loading was found to be 1.0 phr of GO, at a 90:10 ratio of ABS to talc, and 4 phr of SEBS-g-MAH, to obtain maximum mechanical and thermal stability of the composites. The findings of the present study revealed that graphene oxide and SEBS-g-MAH could be utilized as a filler and a compatibilizer respectively, in ABS/talc composites to enhance the mechanical and thermal stability.

## Figures and Tables

**Figure 1 polymers-13-03180-f001:**
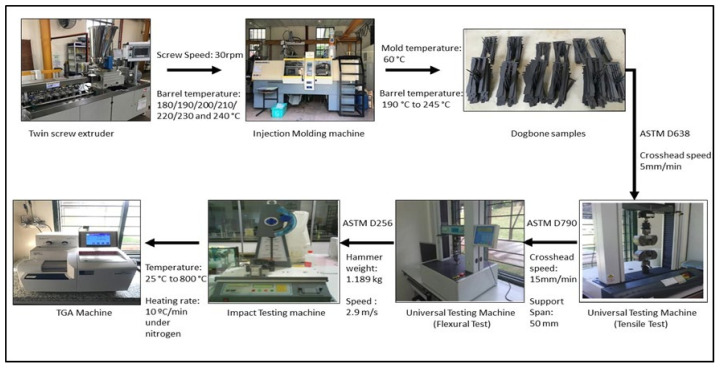
Schematic diagram of the production of ABS/Talc/GO/SEBS-g-MAH through melt blending.

**Figure 2 polymers-13-03180-f002:**
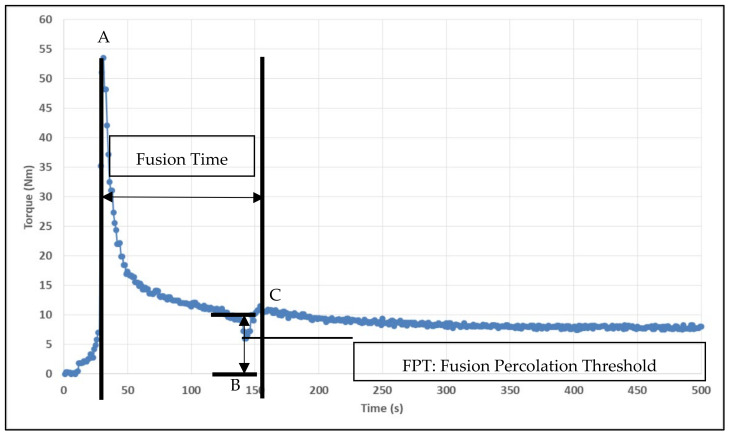
General fusion curve of unfilled ABS compound melted in Haake internal Mixer at a temperature of 190 °C, with the speed of 90 rpm at 15 min of blending time.

**Figure 3 polymers-13-03180-f003:**
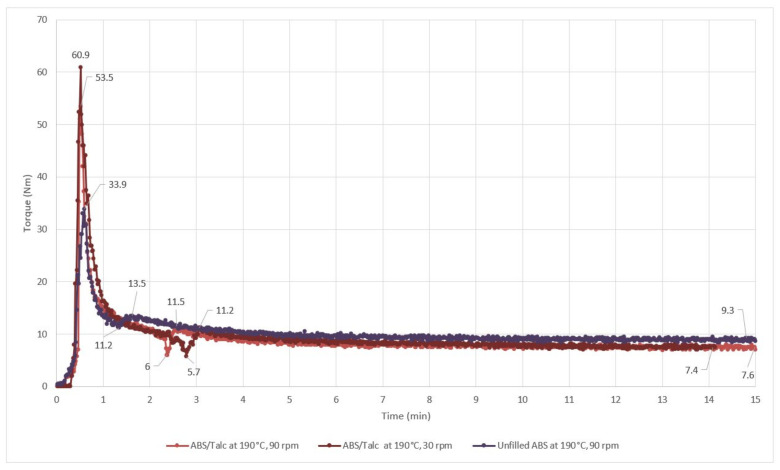
Effect of talc filler on fusion time fusion torque of ABS composites.

**Figure 4 polymers-13-03180-f004:**
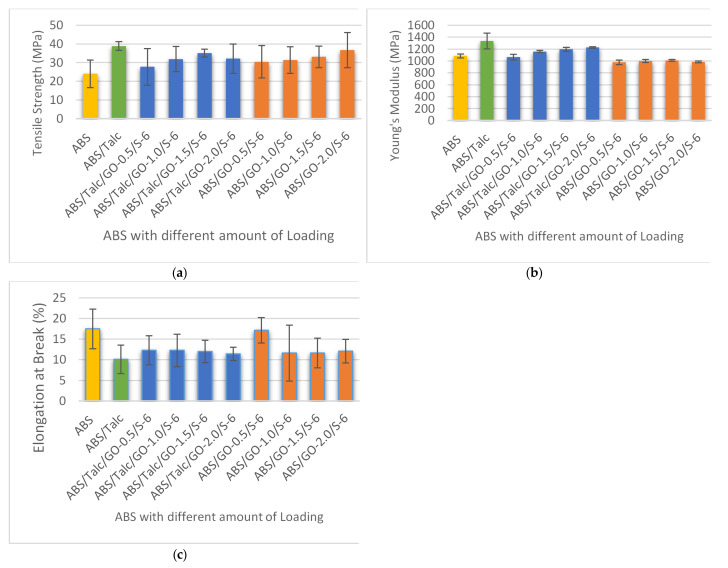
(**a**) Tensile strength, (**b**) Young’s modulus, and (**c**) elongation at break of ABS and ABS/talc, with different loading of GO.

**Figure 5 polymers-13-03180-f005:**
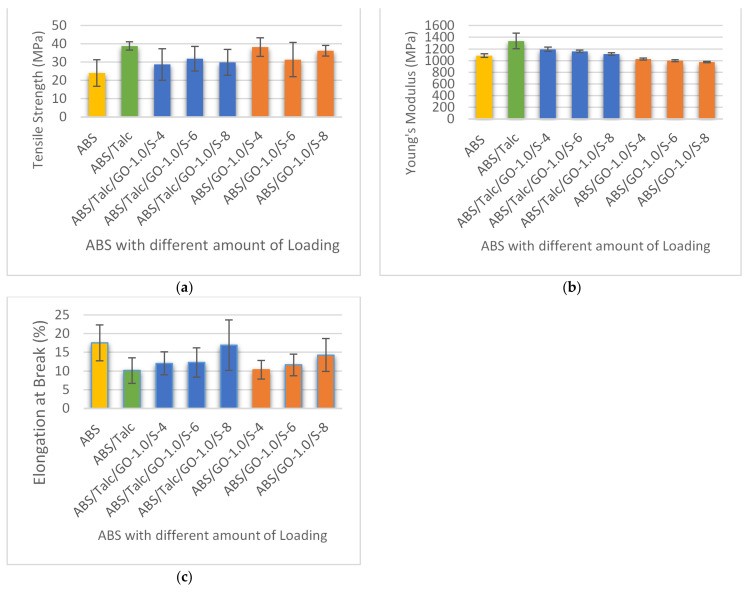
(**a**) Tensile strength, (**b**) Young’s modulus, and (**c**) elongation at break of ABS and ABS/talc, with different loading of SEBS-g-MAH.

**Figure 6 polymers-13-03180-f006:**
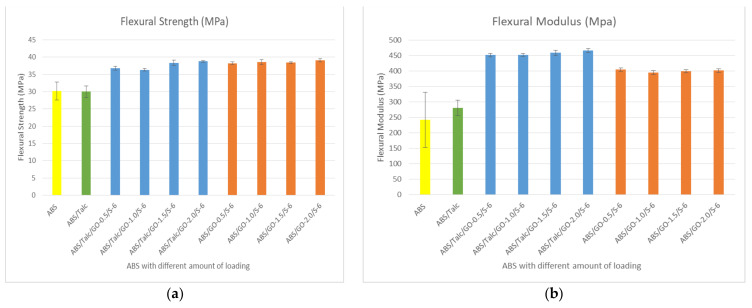
(**a**) Flexural strength and (**b**) flexural modulus of ABS and ABS/talc, with different loading of GO.

**Figure 7 polymers-13-03180-f007:**
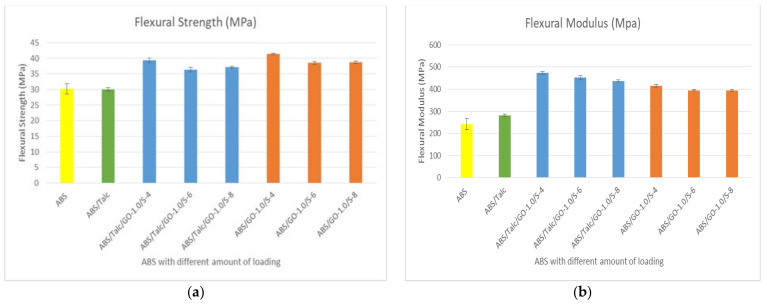
(**a**) Flexural strength and (**b**) flexural modulus of ABS and ABS/talc, with different loading of SEBS-g-MAH.

**Figure 8 polymers-13-03180-f008:**
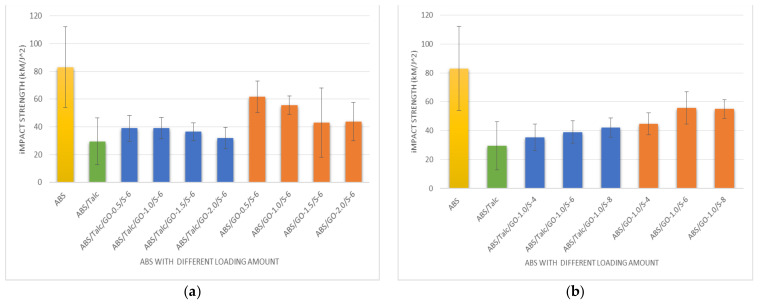
Impact strength of ABS composite with different loading of (**a**) graphene oxide (GO) and (**b**) SEBS-g-MAH.

**Figure 9 polymers-13-03180-f009:**
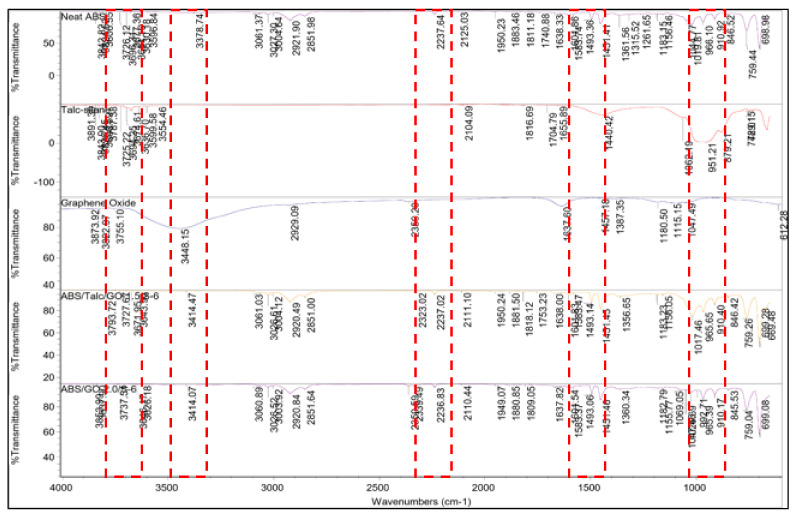
FTIR spectrum of ABS, talc, ABS/talc/GO −1.5/S6, and ABS/GO −2.0/S6.

**Figure 10 polymers-13-03180-f010:**
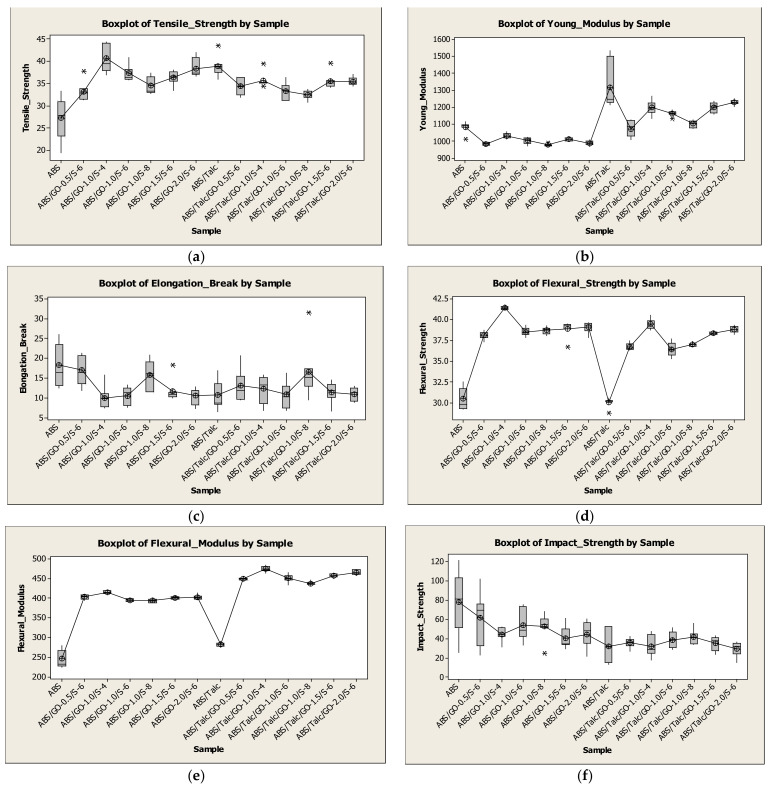
Boxplot for the selected responses using 14 different compositions.

**Table 1 polymers-13-03180-t001:** Various compounding formulations of composites.

NO.	Sample	ABS (%)	Talc (%)	GO (phr)	SEBS-g-MAH (phr)
1	ABS	100	-	-	-
2	ABS/TALC	90	10	-	-
3	ABS/Talc/GO-0.5/S6	90	10	0.50	6
4	ABS/Talc/GO-1.0/S6	90	10	1.00	6
5	ABS/Talc/GO-1.5/S6	90	10	1.50	6
6	ABS/Talc/GO-2.0/S6	90	10	2.00	6
7	ABS/Talc/GO-1.0/S4	90	10	1.00	4
8	ABS/Talc/GO-1.0/S8	90	10	1.00	8
9	ABS/GO-0.5/S6	100	-	0.50	6
10	ABS/GO-1.0/S6	100	-	1.00	6
11	ABS/GO-1.5/S6	100	-	1.50	6
12	ABS/GO-2.0/S6	100	-	2.00	6
13	ABS/GO-1.0/S4	100	-	1.00	4
14	ABS/GO-1.0/S8	100	-	1.00	8

Sample No. 1 ABS acts as a control for these compounding formulations.

**Table 2 polymers-13-03180-t002:** Parameter settings for melt mixing.

Parameter	Value			
Speed	90 rpm	90 rpm	30 rpm	30 rpm
Temperature	190 °C	220 °C	200 °C	190 °C
Duration	15 min	15 min	15 min	15 min

**Table 3 polymers-13-03180-t003:** ABS/talc composites’ composition, fusion torque, and final torque.

Speed (rpm)	ABS (%)	Talc (%)	Fusion Torque (Nm)	Final Torque (Nm)
30	90	10	11.2	7.4
90	90	10	11.5	7.6
90	100	-	13.5	9.3

**Table 4 polymers-13-03180-t004:** ABS/talc composites’ composition and fusion percolation threshold (FPT).

Speed (rpm)	ABS (%)	Talc (%)	FPT (Nm)
30	90	10	5.5
90	90	10	5.5
90	100	-	2.3

**Table 5 polymers-13-03180-t005:** Thermogravimetric analysis (TGA) data of ABS and ABS/talc composites with different loading amounts of GO.

Sample	Decomposition Temperature (Tᴅ) in °C at Different Weight Loss (%)					Residue wt. % (WR) at 800 °C	Inflection Point (°C)
	2	5	10	20	30		
ABS	362.10	384.77	396.61	406.08	411.86	0.98	422.45
ABS/Talc	363.06	380.02	388.43	397.56	403.81	7.76	409.81
ABS/Talc/GO-0.5/S-6	368.57	385.69	394.96	404.96	411.85	8.16	416.94
ABS/Talc/GO-1.0/S-6	366.11	384.86	394.34	403.80	410.38	9.23	413.08
ABS/Talc/GO-1.5/S-6	361.23	384.76	394.78	404.83	411.66	7.60	415.96
ABS/Talc/GO-2.0/S-6	361.79	384.67	395.76	405.60	412.28	8.27	418.33
ABS/GO-0.5/S-6	371.23	384.91	393.23	401.42	407.10	3.41	413.30
ABS/GO-1.0/S-6	371.66	385.02	395.09	405.00	410.10	2.03	423.94
ABS/GO-1.5/S-6	370.42	387.00	397.07	406.87	413.11	2.15	424.93
ABS/GO-2.0/S-6	373.40	389.58	399.27	408.92	415.07	2.29	423.42

**Table 6 polymers-13-03180-t006:** Thermogravimetric analysis (TGA) data of ABS and ABS/talc composites with different loading amounts of SEBS-g-MAH.

Sample	Decomposition Temperature (Tᴅ) in °C at Different Weight Loss (%)					Residue wt. % (WR) at 800 °C	Inflection Point (°C)
	2	5	10	20	30		
ABS	362.10	384.77	396.61	406.08	411.86	0.98	422.45
ABS/Talc	363.06	380.02	388.43	397.56	403.81	7.76	409.81
ABS/Talc/GO-1.0/S-4	365.60	382.60	392.98	401.68	410.45	9.10	412.85
ABS/Talc/GO-1.0/S-6	366.11	384.86	394.34	403.80	410.38	9.23	413.08
ABS/Talc/GO-1.0/S-8	369.51	383.07	392.36	401.50	408.30	9.75	417.01
ABS/GO-1.0/S-4	359.90	386.46	397.48	407.55	413.37	0.47	421.62
ABS/GO-1.0/S-6	362.66	386.69	398.00	408.00	414.10	2.66	422.94
ABS/GO-1.0/S-8	370.69	387.56	398.37	408.66	415.11	2.99	423.57

**Table 7 polymers-13-03180-t007:** The results of ANOVA for the selected variables.

ANOVA						
		Sum of Squares	df	Mean Square	F	Significant (*p*-Value)
TS	Between Groups	987.150	13	75.935	14.725	0.000
	Within Groups	433.169	84	5.157		
	Total	1420.319	97			
YM	Between Groups	1,069,730.138	13	82,286.934	42.979	0.000
	Within Groups	160,826.240	84	1914.598		
	Total	1,230,556.378	97			
EB	Between Groups	741.339	13	57.026	4.045	0.000
	Within Groups	1184.255	84	14.098		
	Total	1925.594	97			
FS	Between Groups	947.920	13	72.917	177.294	0.000
	Within Groups	34.547	84	0.411		
	Total	982.468	97			
FM	Between Groups	402,722.339	13	30,978.641	472.662	0.000
	Within Groups	5505.430	84	65.541		
	Total	408,227.769	97			
IS	Between Groups	16,477.942	13	1267.534	5.519	0.000
	Within Groups	19,292.899	84	229.677		
	Total	35,770.841	97			

## Data Availability

The data presented in this study are available on request from the corresponding author.

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
