# Peer review of "The Effect of Graphene Oxide and SEBS-g-MAH Compatibilizer on Mechanical and Thermal Properties of Acrylonitrile-Butadiene-Styrene/Talc Composite"

_polymers, 2021, doi:10.3390/polym13183180_

Round 1

Reviewer 1 Report

This paper "The effect of graphene oxide and sebs-g-mah compatibilizer on mechanical and thermal properties of acrylonitrile-butadiene-styrene/talc composite" is well written. I recommend it to publish on this journal. But before publish, there are several comments need to be addressed:

  • The corresponding SEM images should be used to demonstrate the addition of GO and SEBS-g-MAH improved interfacial compatibility.
  • Explaining “the exothermal energy from the decomposition of GO could enhance the decomposition temperature of ABS matrix”.
  • Why the residue at 800of ABS/Talc/GO-1.0/ SEBS-g-MAH decreased with the SEBS-g-MAH content increasing? And the reason of the variation of the residue at 800of ABS/Talc/GO/S-6 and ABS/GO/S-6?
  • In table 0, why the thermal stability of composites with talc is lower than without talc.
  • In Fig. 3, why the tensile strength of composites showed a trend of increasing first and then decreasing with GO content increasing?

Author Response

Comments and Suggestions for Authors:

This paper "The effect of graphene oxide and sebs-g-mah compatibilizer on mechanical and thermal properties of acrylonitrile-butadiene-styrene/talc composite" is well written. I recommend it to publish on this journal.

Response: Thanks for review the manuscript and provide valuable comments and suggestions to improve the manuscript. The manuscript is revised following the reviewer's comments and suggestions.

But before publish, there are several comments need to be addressed:

(1) The corresponding SEM images should be used to demonstrate the addition of GO and SEBS-g-MAH improved interfacial compatibility.

Response: Thanks for the suggestion. It is obvious that the inclusion of SEM images will demonstrate the improvement of interfacial compatibility of GO and SEBS-g-MAH in composite. However, we are unable to provide the SEM images due to current movement control order. Our laboratory is closed and we are unable to conduct any experiments. However, the aim of the present study to determine the influence graphene oxide and sebs-g-mah compatibilizer on mechanical and thermal properties of the isolated composite. The characterizations on mechanical and thermal properties showed that the inclusion of GO and SEBS-g-MAH are potentiality improved the mechanical and thrmal properties of the composites.

However, this is our ongoing project. We will conduct these experiments and we will report in our next paper.

(2) Explaining “the exothermal energy from the decomposition of GO could enhance the decomposition temperature of ABS matrix”.

Response: Added explanation.  

(3) Why the residue at 800of ABS/Talc/GO-1.0/ SEBS-g-MAH decreased with the SEBS-g-MAH content increasing? And the reason of the variation of the residue at 800of ABS/Talc/GO/S-6 and ABS/GO/S-6?

Response: Data from Table 5 and 6 was revised and we found out some error in reporting. The triplicate values have been revised and confirmed.

(4) In table 0, why the thermal stability of composites with talc is lower than without talc.

Response:  Generally Talc is the softest mineral filler with its high resistivity, low gas permeability and high lubricity. However, the decrease of thermal stability of composites with talc was might due to the poor dispersion in ABS matrix along with void formation.

(5) In Fig. 3, why the tensile strength of composites showed a trend of increasing first and then decreasing with GO content increasing?

Response: The explanation has been added in manuscript. The addition of GO improves the interfacial adhesion between the matrix and fillers particles and therefore enhancing the tensile strength of the composite. However, the decrease of the tensile strength with increasing GO content might due to the saturation of GO in ABS matrix.

Reviewer 2 Report

  1. The manuscript is well organized, and the subject is well presented. The methods used are sound and the presentation and discussion of results is logical.
  2. The manuscript requires some major revisions to bring it to a level worthy of publication. My recommendations are detailed below:
  3. The current study investigates the mechanical and thermal characteristics of acrylonitrile-butadienestyrene/talc composite due to adding graphene oxide and sebs-g-mah compatibilizer. For this, the authors prepared two types of composites and add graphene oxide powder in different phr levels. The authors reported on improved mechanical and thermal properties. The authors also study the fusion characteristics through fusion melting process. The authors reported % increase in mechanical properties but it is not clear what are these properties which are TS, FS and FM?
  4. First of all, the authors should mention the full wording or the acronyms such as TF, FS and FM first time they appear in the manuscript.
  5. Please remove this sentence from the abstract “It is known that incorporation of talc reduces the stiffness of composites.” Or say you added talc to reduce the stiffness in composites.
  6. The abstract needs further improvement, Please consider reviewing the abstract and highlight the novelty, major findings and conclusions.
  7. Before this sentence, “The objective of this study is to investigation”, the authors are encouraged to answer the following question: What is the research gap did you find from the previous researchers in your field? Mention it properly. It will improve the strength of the article.
  8. In the materials and methods sections, the authors should add any images for test equipment, test setup or fabricated samples and test machines used in the current work. This is an experimental study and more graphical details are needed to better showcase your work.
  9. “The three-point bending flexural test will be performed according to the ASTM D790” fix this sentence it should be “The three point bending flexural test were performed according to the ASTM D790”
  10. “The average of 10 specimens results in flexural strength (FS) and flexural” you already said before you tested 10 samples for each of the mechanical tests so there is no need to repeat it.
  11. “The result of impact strength (IM) calculated was the average of ten specimens.” Again you mentioned it here….
  12. I think the manuscript requires extensive English editing and rephrasing. There is a lot of repetitions in some sections which can be removed.
  13. “The infra-red spectroscopy analyses were carried out” was carried out
  14. “However, the experiment was unsuccessful as talc did not mixed with ABS” check this sentence for English editing.
  15. “However, the experiment was unsuccessful as talc did not mixed with ABS” any idea of why this problem happened?
  16. “The analysis was supported with Jang (Polli et al., 2009) that claimed the normal ABS degrades at 410°C with residue of 1% above 600°C.” this sentence needs to be checked for editing and rephrasing. Perhaps you should say “The analysis was supported by the results reported the work of Jang (Polli et al., 2009) which claimed the normal ABS degrades at 410°C with residue of 1% above 600°C.”
  17. Why figure 1 is repeated twice?
  18. Figure 1. Fusion curve of ABS/Talc at 190°C and 90rpm.
  19. Figure 1. General fusion curve of unfilled ABS compound melted in Haake internal Mixer at a temperature of 190°C with speed of 90rpm at 15 minutes of blending time.
  20. “Figure 1. General fusion curve of unfilled ABS compound melted in Haake internal Mixer at a temperature of 190°C with speed of 90rpm at 15 minutes of blending time.” This figure is not clear, I am struggling to see points B and C on the graph, please enlarge that section in the graph so we could clearly see what you are talking about in there.
  21. “2 times longer than the unfilled PVC.” Use the word two not the number 2
  22. “This may be due to higher melt viscosity of the composite” please support this claim with a reference, or mention what past studies found regarding this matter.
  23. “Consequently, higher shear and heat transfer not only offers increment in fusion temperature but also reduction of melt viscosity.” Is this a fact, if yes then please support with reference(s).
  24. “This is due to the low friction energy generated between the ABS particles with talc and the metal surface.” Have you measured the friction energy or is this a speculation?
  25. “which indicates that talc filled ABS needed more thermal energy to be fused together” rephrase this sentence, for example instead of using the word needed, say required and instead of “more thermal energy” say higher thermal energy.
  26. “From the findings, it is expected that the addition of GO will improve the interfacial adhesion between the matrix and fillers particles.” Why? Is this according to past literature/knowledge in the field then please support this with references and explain further.
  27. “Hence, enhancing the mechanical properties of the composite. In order to study the tensile properties of ABS composites, the test was done according to ASTM D638.” You already said this in the materials and methods sections why mention it again here?
  28. Combine figures 3 and 4 and 5 into one figure and use a and b and c instead.
  29. Figure 4 and 5 please fix the y axis and remove the 0.000 from the axis values.
  30. “The bonding promotes a strong interfacial adhesion between filler and matrix.” Is this an explanation or just a fact, it is not clear if this statements is related to the work results or not. Also please reference this fact unless you found it on your own and not from the open literature.
  31. “This will promotes a better interac” remove the s from promotes.
  32. Combine figures 6-8 into one larger paragraph is possible.
  33. “This inhibition occur because of strong interaction between polymer and matrix” a reference is needed here.
  34. Conclusions is weak and must be enhanced significantly, a suggestion is the authors add 1-2 bullet points from each subsection in the results and discussion section.
  35. The authors are encouraged to include more discussion in sections 3.4/3.5 and 3.6 and critically discuss the observations from this investigation with existing literature.
  36. Overall the study is worth publishing but the authors must improve the writing style of the manuscript.

Author Response

Comments and Suggestions for Authors:

 (1) The manuscript is well organized, and the subject is well presented. The methods used are sound and the presentation and discussion of results is logical.

Response: Thanks for review the manuscript and provide valuable comments and suggestions to improve the manuscript. The manuscript is revised following the reviewer's comments and suggestions.

 (2) The manuscript requires some major revisions to bring it to a level worthy of publication. My recommendations are detailed below:

Response: Thanks for proving valuable comments and suggestions. We have amended the manuscript following the reviewer comments.

(3) The current study investigates the mechanical and thermal characteristics of acrylonitrile-butadienestyrene/talc composite due to adding graphene oxide and sebs-g-mah compatibilizer. For this, the authors prepared two types of composites and add graphene oxide powder in different phr levels. The authors reported on improved mechanical and thermal properties. The authors also study the fusion characteristics through fusion melting process. The authors reported % increase in mechanical properties but it is not clear what are these properties which are TS, FS and FM?

Response: Revised. TS, FS and FM  are mechanical properties such as tensile strength (TS), flexural strength (FS) and flexural modulus (FM).

(4) First of all, the authors should mention the full wording or the acronyms such as TF, FS and FM first time they appear in the manuscript.

Response:   Added.

(5)  Please remove this sentence from the abstract “It is known that incorporation of talc reduces the stiffness of composites.” Or say you added talc to reduce the stiffness in composites.

Response: Corrected as suggested.

(6) The abstract needs further improvement, Please consider reviewing the abstract and highlight the novelty, major findings and conclusions.

Response: The abstract has been revised.

(7) Before this sentence, “The objective of this study is to investigation”, the authors are encouraged to answer the following question: What is the research gap did you find from the previous researchers in your field? Mention it properly. It will improve the strength of the article.

Response: Revised and amended as suggested.

(8) In the materials and methods sections, the authors should add any images for test equipment, test setup or fabricated samples and test machines used in the current work. This is an experimental study and more graphical details are needed to better showcase your work.

Response: Added Figure 1 “Schematic diagram of the production of ABS/Talc/GO/SEBS-g-MAH through melt blending”.

(9) “The three-point bending flexural test will be performed according to the ASTM D790” fix this sentence it should be “The three point bending flexural test were performed according to the ASTM D790”

Response:   Corrected.

(10) “The average of 10 specimens results in flexural strength (FS) and flexural” you already said before you tested 10 samples for each of the mechanical tests so there is no need to repeat it.

Response: Deleted.

(11) “The result of impact strength (IM) calculated was the average of ten specimens.” Again you mentioned it here….

Response: Deleted.  

(12) I think the manuscript requires extensive English editing and rephrasing. There is a lot of repetitions in some sections which can be removed.

Response: Revised the English language of the manuscript.

(13) “The infra-red spectroscopy analyses were carried out” was carried out

Response: Corrected.

(14) “However, the experiment was unsuccessful as talc did not mixed with ABS” check this sentence for English editing.

Response: Revised as advised. The statement can be found from line 276 until 277.

(15) “However, the experiment was unsuccessful as talc did not mixed with ABS” any idea of why this problem happened?

Response: Revised as advised. The statement can be found from line 318 until 323.

(16) “The analysis was supported with Jang (Polli et al., 2009) that claimed the normal ABS degrades at 410°C with residue of 1% above 600°C.” this sentence needs to be checked for editing and rephrasing. Perhaps you should say “The analysis was supported by the results reported the work of Jang (Polli et al., 2009) which claimed the normal ABS degrades at 410°C with residue of 1% above 600°C.”

Response: Revised.

(17) Why figure 1 is repeated twice?

Response: Corrected.

(18) Figure 1. Fusion curve of ABS/Talc at 190°C and 90rpm.

Response: Revised.

(19) Figure 1. General fusion curve of unfilled ABS compound melted in Haake internal Mixer at a temperature of 190°C with speed of 90rpm at 15 minutes of blending time.

Response: Corrected.

(20) “Figure 1. General fusion curve of unfilled ABS compound melted in Haake internal Mixer at a temperature of 190°C with speed of 90rpm at 15 minutes of blending time.” This figure is not clear, I am struggling to see points B and C on the graph, please enlarge that section in the graph so we could clearly see what you are talking about in there.

Response: Replaced with clear figure.

(21) “2 times longer than the unfilled PVC.” Use the word two not the number 2

Response: Revised.

(22) “This may be due to higher melt viscosity of the composite” please support this claim with a reference, or mention what past studies found regarding this matter

Response:   Added reference

(23)  “Consequently, higher shear and heat transfer not only offers increment in fusion temperature but also reduction of melt viscosity.” Is this a fact, if yes then please support with reference(s).

Response: Added.

(24) “This is due to the low friction energy generated between the ABS particles with talc and the metal surface.” Have you measured the friction energy or is this a speculation?

Response: Deleted these lines.

(25) “which indicates that talc filled ABS needed more thermal energy to be fused together” rephrase this sentence, for example instead of using the word needed, say required and instead of “more thermal energy” say higher thermal energy.

Response: Revised as advised.

(26) “From the findings, it is expected that the addition of GO will improve the interfacial adhesion between the matrix and fillers particles.” Why? Is this according to past literature/knowledge in the field then please support this with references and explain further.

Response: Revised as advised.

(27) “Hence, enhancing the mechanical properties of the composite. In order to study the tensile properties of ABS composites, the test was done according to ASTM D638.” You already said this in the materials and methods sections why mention it again here?

Response:   Deleted.

(28) Combine figures 3 and 4 and 5 into one figure and use a and b and c instead.

Response: Combined.

(29) Figure 4 and 5 please fix the y axis and remove the 0.000 from the axis values.

Response: Revised as suggested.

(30) “The bonding promotes a strong interfacial adhesion between filler and matrix.” Is this an explanation or just a fact, it is not clear if this statements is related to the work results or not. Also please reference this fact unless you found it on your own and not from the open literature.

Response:  Revised. Yes, this is a fact based on the finding related to the work.

(31) “This will promotes a better interac” remove the s from promotes

Response: Removed.

(32) Combine figures 6-8 into one larger paragraph is possible.

Response: Combined as suggested.

(33) “This inhibition occur because of strong interaction between polymer and matrix” a reference is needed here.

Response: Added reference  

(34) Conclusions is weak and must be enhanced significantly, a suggestion is the authors add 1-2 bullet points from each subsection in the results and discussion section.

Response:  Revised the conclusion.

(35) The authors are encouraged to include more discussion in sections 3.4/3.5 and 3.6 and critically discuss the observations from this investigation with existing literature.

Response: Corrected. Additional discussion for section 3.5 for flexural properties for effect of loading of GO from line 472 until 499, section 3.6 for flexural properties for effect of loading of SEBS-g-MAH from line 501 until 514, Section 3.7 for impact properties of loading of GO and SEBS-g-MAH from line 516 until 529.  Discussion on 3.6 has been deliberated as reviewed.

(36) Overall the study is worth publishing but the authors must improve the writing style of the manuscript.

Response:  Thanks for the comments. We have revised English language of the manuscript.

Round 2

Reviewer 2 Report

All questions answered and paper can be accepted